# History Filtering in Imperfect Information Games: Algorithms and Complexity

**Christopher Solinas**[1], **Douglas Rebstock**[1], **Nathan R. Sturtevant**[1,2], **Michael Buro**[1]

[1]Department of Computing Science, University of Alberta
[2]Alberta Machine Intelligence Institute (Amii)
{solinas,drebstoc,nathanst,mburo}@ualberta.ca

## Abstract

Historically applied exclusively to perfect information games, depth-limited search with value functions has been key to recent advances in AI for imperfect information games. Most prominent approaches with strong theoretical guarantees require *subgame decomposition*—a process in which a subgame is computed from public information and player beliefs. However, subgame decomposition can itself require non-trivial computations, and its tractability depends on the existence of efficient algorithms for either full enumeration or generation of the histories that form the root of the subgame. Despite this, no formal analysis of the tractability of such computations has been established in prior work, and application domains have often consisted of games, such as poker, for which enumeration is trivial on modern hardware.

Applying these ideas to more complex domains requires understanding their cost. In this work, we introduce and analyze the computational aspects and tractability of filtering histories for subgame decomposition. We show that constructing a single history from the root of the subgame is generally intractable, and then provide a necessary and sufficient condition for efficient enumeration. We also introduce a novel Markov Chain Monte Carlo-based generation algorithm for trick-taking card games—a domain where enumeration is often prohibitively expensive. Our experiments demonstrate its improved scalability in the trick-taking card game *Oh Hell*. These contributions clarify when and how depth-limited search via subgame decomposition can be an effective tool for sequential decision-making in imperfect information settings.

## 1 Introduction

Games are a standard model for sequential decision-making. As the number of sequential decisions needed to play increases, the size of the game's state space can grow exponentially—quickly becoming too large to search exhaustively. Depth-limited search navigates this issue by replacing decision points below a certain depth with a value function that captures or approximates the value of playing the subgame from that position onward. This can massively boost scalability and has been key to several famous results in perfect information games including Checkers (Schaeffer and Lake [1996]), Chess (Campbell *et al.* [2002]) and Go (Silver *et al.* [2017]).

The same idea has recently been successfully applied to certain imperfect information games such as poker (Brown and Sandholm [2019]; Moravčík *et al.* [2017])—achieving similarly impressive results by defeating human experts. In contrast to the perfect information setting, for which computing the value function only requires evaluating the current history, most successful depth-limited search algorithms in imperfect information games require evaluating whole sets of histories and their reach probabilities. This information is used to generalize the concept of a subgame and its value

37th Conference on Neural Information Processing Systems (NeurIPS 2023).

by decomposing the game tree into *public belief states*, which are analogous to belief states in Markov systems such as Hidden Markov Models (HMMs) and Partially Observable Markov Decision Processes (POMDPs). This approach is commonly referred to as *subgame decomposition*; we describe the associated history and reach probability computations as *history filtering*. Although current theory establishes the necessary information for theoretically sound depth-limited search in imperfect information games (Kovařík *et al.* [2020]), it is unclear how, or if, this information can be computed efficiently. Understanding these computations provides insight into scaling search to larger, more complex imperfect information games.

In this work, we define variants of history filtering for subgame decomposition that are useful for search—called *enumeration* and *generation*—while also developing a suitable notion for efficiently solving them. Efficient solutions should take at most polynomially many steps in the length of the input observation sequence. We show that, in general, such algorithms only exist if P = NP. From there, we investigate methods for efficient enumeration and generation. First, by identifying a structural property of the game tree that is both a necessary and sufficient condition for efficient enumeration, and then by introducing a novel, unbiased generation algorithm for trick-taking card games, based on Markov Chain Monte Carlo. Through experiments in *Oh Hell*, we validate its improved scalability and highlight the potential advantages of this approach. Our contributions advance the theory of depth-limited search in imperfect information domains.

## 2 Background

In this section, we summarize the concepts and algorithms related to history filtering and public belief states.

### 2.1 Factored Observation Stochastic Games

The recent introduction of Factored Observation Stochastic Games (FOSGs) (Kovařík *et al.* [2019]) has helped clarify fundamental concepts about decomposing public and private observations and information in multi-agent, partially-observable sequential decision problems.

An FOSG is a tuple $G = \langle \mathcal{N}, \mathcal{W}, \mathcal{P}, w^0, \mathcal{A}, \mathcal{T}, \mathcal{R}, \mathcal{O} \rangle$. $\mathcal{N} = \{1, ..., N\}$ represents the set of players, $\mathcal{W}$ is the set of world states, and $\mathcal{A}$ is the set of joint actions. $\mathcal{P} : \mathcal{W} \to 2^{\mathcal{N}}$ is the player function, which describes which players act in which world states. $\mathcal{T} : \mathcal{W} \times \mathcal{A} \to \Delta\mathcal{W}$ is the state transition function, where $\Delta\mathcal{W}$ represents the set of probability distributions over $\mathcal{W}$. $\mathcal{R} : \mathcal{W} \times \mathcal{A} \to \mathbb{R}^N$ assigns a reward to each player, and $\mathcal{O} : \mathcal{W} \times \mathcal{A} \times \mathcal{W} \to \mathbb{O}^{N+1}$ is the observation function—which maps transitions (world state-action-world state triples) to private observations for each player and a public observation common to all players.

Games start at the initial world state $w^0$. In any world state $w \in \mathcal{W}$, player $i$ acts when $i \in \mathcal{P}(w)$. The joint action set $\mathcal{A} := \prod_{i \in \mathcal{N}} \mathcal{A}_i$ is defined as the product of each player's individual action sets across all $w \in \mathcal{W}$. $\mathcal{A}_i(w) \subset \mathcal{A}_i$ denotes the legal actions for $i$ in $w$, and $\mathcal{A}(w) := \prod_{i \in \mathcal{P}(w)} \mathcal{A}_i(w)$ is the set of legal joint actions in $w$. Play proceeds when each $i \in \mathcal{P}(w)$ chooses an action $a_i \in \mathcal{A}_i(w)$—resulting in joint action $a := (a_i)_{i \in \mathcal{P}(w)}, a \in \mathcal{A}(w)$. The next state $w'$ is sampled from $\mathcal{T}(w, a)$, while the reward is determined by evaluating $\mathcal{R}(w, a)$. Finally, $\mathcal{O}(w, a, w')$ is factored into public and private observations as $(\mathcal{O}_{\text{priv}(1)}(w, a, w'), ..., \mathcal{O}_{\text{priv}(N)}(w, a, w'), \mathcal{O}_{\text{pub}}(w, a, w'))$.

### 2.2 Policies, Reach Probabilities, and Beliefs

A **history** is a sequence $h := (w^0, a^0, w^1, a^1, ..., w^t)$ of world states and actions for which $w^k \in \mathcal{W}, a^k \in \mathcal{A}(w^k)$, and $\mathbb{P}[\mathcal{T}(w^k, a^k) = w^{k+1}] > 0$ for $k, 0 \leq k \leq t - 1$. We refer to $|h| := t$ as the *length* of $h$. The set of all legal histories is denoted as $\mathcal{H}$. We use the standard notation $h' \sqsubseteq h$ to denote that $h'$ is a **prefix history** of $h$ (i.e., $h'$ is a subsequence of $h$ starting at $w^0$ and ending in a world state). A **terminal history** $z \in \mathcal{Z}$ signifies that play has reached a world state where the game ends. The **utility** for player $i$, $u_i : \mathcal{Z} \to \mathbb{R}$ is the sum of all rewards accumulated by $i$ over the world state and action sequence.

A history $h$ produces a **public state**, which is the sequence of public observations $s_{pub}(h) := (O_{pub}^1, O_{pub}^2, ..., O_{pub}^t)$ produced along the trajectory with $O_{pub}^k := \mathcal{O}_{pub}(w^{k-1}, a^{k-1}, w^k)$. We

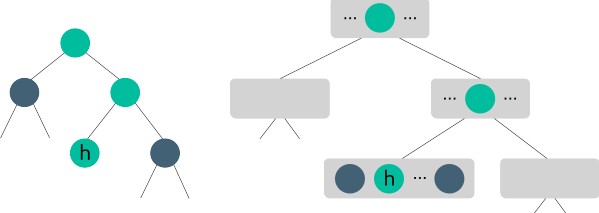

Figure 1: The same history viewed in the world state tree (left) and the public tree (right). Grey boxes are public states, which can be consistent with many histories from the world state tree.

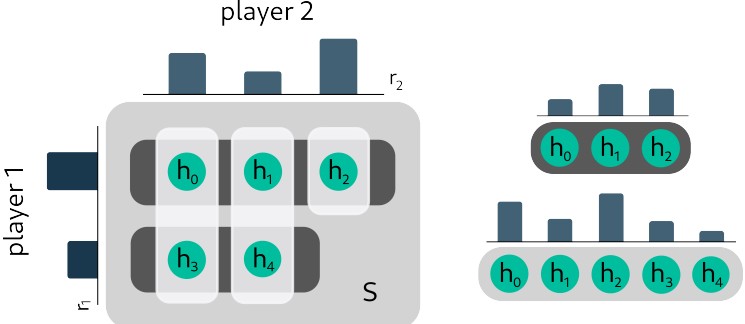

Figure 2: (Left) An example public belief state in a two-player game with ranges $r_1$ and $r_2$. $S$ is consistent with a set of 5 histories. Player 1's range contains two private information states. (Right) A player's beliefs (top) and a joint range (bottom). Player 1's beliefs, given the first private infostate, consist of reach probabilities for the individual histories in the infostate. The joint range consists of reach probabilities over all histories in $S$.

denote a public state generated by an unknown history as $S \in \mathcal{S}$, where $\mathcal{S}$ is the set of all public states (and the vertex set of the **public tree**, see Figure 1). Likewise, player $i$'s **private information state** $S_i \in \mathcal{S}_i$ similarly captures the sequence of observations seen only by that player: $s_i(h) := (O_i^1, O_i^2, ..., O_i^t)$, where $O_i^k := \mathcal{O}_{priv(i)}(w^{k-1}, a^{k-1}, w^k)$. Taken together, $(s_{pub}(h), s_i(h))$ represents all information available to player $i$ at history $h$, and is referred to as the player's **information state** or **infostate**. $\mathcal{S}_i(S)$ denotes the set of all infostates consistent with public state $S$ and is a partition of $S$. We use $\mathcal{H}_S := \{h \in \mathcal{H} : s_{pub}(h) = S\}$ to denote the set of histories consistent with public state $S$. Since all players receive (possibly empty) observations at every world state transition, this formalism avoids *non-timeability* and *thick infostates* present in the extensive form. See Kovařík *et al.* [2020] for details.

A player plays according to a **policy** $\pi_i : \mathcal{S}_i \to \Delta(\mathcal{A})$ which maps player $i$'s infostates to the set of probability distributions over action set $\mathcal{A}$. A **joint policy** $\pi = (\pi_1, ..., \pi_n)$ is a tuple consisting of every player's policy. The **reach probability** of a history under $\pi$ is $P^\pi(h) = P_c(h)P_1^\pi(h)P_2^\pi(h)...P_N^\pi(h)$ where each $P_i^\pi(h)$ is a product of action probabilities taken by player $i$ to reach $h$ from $w^0$, and $P_c(h)$ is the product of all probabilities from chance transitions taken according to the stochastic transition function $\mathcal{T}$. The reach probability of an infostate (or public state) $S_i$ under $\pi$ can be expressed as $P^\pi(S_i) := \sum_{h \in S_i} P^\pi(h)$. Infostate reach probabilities can also be decomposed into $P^\pi(S_i) = P_i^\pi(S_i)P_{-i}^\pi(S_i)$ where $-i$ denotes all players except $i$. Since all $h \in S_i$ are indistinguishable to $i$, $P_i^\pi(S_i) = P_i^\pi(h)$, whereas $P_{-i}^\pi(S_i) = \sum_{h \in \mathcal{H}_{S_i}} P_{-i}^\pi(h)$ (Kovařík *et al.* [2020]).

Kovařík *et al.* [2020] define a **belief** under joint policy $\pi$ as the probability of reaching a history $h$ given the current infostate $S_i$: for any $h \in \mathcal{H}_{S_i}$, $P^\pi(h|S_i) = P^\pi(h)/\sum_{h' \in \mathcal{H}_{S_i}} P^\pi(h')$. The **range** for a joint policy $\pi$ at public state $S$, $r^\pi(S) := ((P_j^\pi(S_i))_{S_i \in \mathcal{S}_j(s_{pub})})_{j=1,2,...,N}$ contains each player's reach probabilities for their infostate partition of $S$. The (normalized) **joint range** $P^\pi(h|S)$ contains the normalized reach probabilities of all $h \in S$. These concepts are visualized in an abstract game in Figure 2. Our analysis focuses on history filtering with respect to the joint range—assuming the joint policy is constant and known by all players.

## 2.3 Depth-Limited Search, Subgames, and Belief State Computation

Several depth-limited search algorithms, such as Information Set Monte Carlo Tree Search (ISMCTS) (Cowling *et al.* [2012]), Player of Games (Schmid *et al.* [2021]), and ReBel (Brown *et al.* [2020]), require methods for simulating actions and state transitions. In these algorithms, search starts from a *ground truth* history—a plausible state of the game—and proceeds until a depth limit; *value functions* represent the players' expected reward in the *subgame* below.

Subgames and their value functions are intuitive in the perfect information setting because they can be rooted at any non-terminal history. However, the necessary information for computing value functions in the imperfect information setting is structured differently. **Subgames** are instead rooted at **public belief states** (PBS), which are tuples $\beta := (S, r^\pi(S))$ composed of a public state and a range. Searching from $\beta$ requires computing histories from $\mathcal{H}_S$ because the actions and transitions necessary for forward simulation are defined for world states, not public states, and computing optimal value functions at the subgame leaves requires ranges defined over all histories in $\mathcal{H}_S$ (Kovařík *et al.* [2020]). Player of Games (Schmid *et al.* [2021]) opts for the Monte Carlo approach, which generates $h \in \mathcal{H}_S$ for player $i$ by sampling an infostate from $S_i \in \mathcal{S}_i(S)$ according to $i$'s component of $r^\pi(S)$, and then sampling according to their beliefs $P^\pi(h|S_i)$ (see Figure 2 for the distinction). ISMCTS (Cowling *et al.* [2012]) is similar, but instead samples directly from the normalized joint range $P^\pi(h|S)$. Either case depends on solving a common computational problem: histories from $\mathcal{H}_S$ must be sampled according to $r^\pi(S)$, given only $S$ and $\pi$.

Belief state computation is an important topic in general Markov systems with partial observability. As such, it appears frequently in areas such as stochastic control (Nayyar *et al.* [2013]), learning in decentralized POMDPs (Dibangoye *et al.* [2016]; Oliehoek [2013]), and multi-agent reinforcement learning (Fickinger *et al.* [2021]). In FOSGs, which are a generalization of POMDPs, search techniques that use public belief states have also been crucial to the development of superhuman poker AI (Brown and Sandholm [2019]; Brown *et al.* [2020]; Moravčík *et al.* [2017]; Schmid *et al.* [2021]). In card game AI, Perfect Information Monte Carlo (PIMC) search was successfully applied to Contract Bridge (Levy [1989]; Ginsberg [2001]) and later Skat (Buro *et al.* [2009]). Both applications rely on game-specific implementations for history filtering. Richards and Amir [2012] described the problem as *information set generation* and provided the first generic, but exponential-time, algorithm for solving it. Seitz *et al.* [2021] provided an algorithm for approximating information set generation based on deep neural networks. Šustr *et al.* [2021] identified that exponential-sized $\mathcal{H}_S$ cause memory issues in methods that explicitly represent the range, and that efficient search requires the range to be represented compactly.

Recently, the idea of *fine-tuning* has helped push the scalability of planning in POMDPs via reinforcement learning (Fickinger *et al.* [2021]) and search in FOSGs (Sokota *et al.* [2021]) by eliminating the need to explicitly represent the range. In reinforcement learning, fine-tuning consists of online updates to a parameterized model of a blueprint policy or Q-value function using sample trajectories with a fixed horizon. These local improvements to the model helped train policies which achieved a new state-of-the-art in self-play Hanabi (Fickinger *et al.* [2021]). Belief fine-tuning (BFT) (Sokota *et al.* [2021]) extended the idea to PBS approximation without pre-training the generative model as a function of the joint policy $\pi$.

In some domains, it is unclear how to structure a dynamics model for fine-tuning such that only legal sample histories are produced. For example, in trick-taking card games, the public observation sequence could indicate that certain players cannot possibly have certain cards in their hands; all histories output by the model should satisfy this constraint. Our approach uses only the dynamics model defined by the game. It is parameter-free and instead calls for simple domain-specific algorithms for history construction and neighbor generation in a Markov chain. This results in unbiased history generation and guarantees that any sample history is legal according to the game rules.

## 3 History Computation in Public Belief States

Our first contribution is to introduce a family of history filtering relations $\texttt{FILTER}(G, \pi)$ which formalize this computation for a given FOSG $G$ and joint policy $\pi$. For simplicity, we limit our

analysis to FOSGs with finite world state sets $\mathcal{W}$ and action sets $\mathcal{A}$, and joint policies that can be evaluated in polynomial time with respect to history length.

## 3.1 Family of Filtering Relations

Computing histories from a given public state is a special case of *particle filtering*, so we name our family of relations FILTER.

**Definition 1.** *(FILTER) For any FOSG $G$ with finite $\mathcal{W}, \mathcal{A}$ and joint policy $\pi$, let FILTER$(G, \pi) :=$ $\{(S, h) \in \Sigma^* \times \Sigma^* : s_{pub}(h) = S, P^\pi(h|S) > 0\}$.*

An FOSG $G$ and a joint policy parameterize the relation—which pairs public states $S \in \mathcal{S}$ and histories consistent with $S$. $S$ is treated as the problem input (encoded using alphabet $\Sigma$) and reachable histories from $\mathcal{H}_S$ are valid outputs.

Our definition treats the FOSG $G$ and the joint policy $\pi$ as fixed and assumes that the FOSG functions ($\mathcal{T}, \mathcal{A}$, etc.) and the policy can be evaluated in polynomial time with respect to their inputs, and are not encoded as inputs to the problem. Thus, the input and output sizes of a problem instance are the lengths of the encoded observation sequence and history, respectively.

A relation $R \subset \Sigma^* \times \Sigma^*$ over alphabet $\Sigma$ is **polynomially balanced** if there exists a polynomial $p$ such that for all $(x, y) \in R, |y| \leq p(|x|)$ (i.e., the length of output $y$ is at most polynomial in the input length) (Jerrum *et al.* [1986]). $R$ is **polynomial-time verifiable** if the predicate $(x, y) \in R$ ($xRy$ for short) can be tested in polynomial time. Lemma 1 states that this is the case for FILTER$(G, \pi)$ with finite $\mathcal{W}$ and $\mathcal{A}$. Finite $\mathcal{W}$ and $\mathcal{A}$ imply polynomial balance and polynomial-time verification is done using $|h|$ evaluations of the observation function and policy to check if $S$ is produced and the reach probability is nonzero. All proofs are in the appendix.

**Lemma 1.** *For any FOSG $G$ with finite $\mathcal{W}, \mathcal{A}$ and arbitrary joint policy $\pi$, FILTER$(G, \pi)$ is polynomially balanced and polynomial-time verifiable.*

## 3.2 Computational Problem Variants

For a binary relation $R$, there are several naturally associated computational problems. Given a problem instance $x \in \Sigma^*$, some of these are:

1. **Existence**: Is there a $y \in \Sigma^*$ such that $xRy$?
2. **Construction**: Return a $y \in \Sigma^*$ such that $xRy$ if one exists.
3. **Generation**: Generate a $y \in \Sigma^*$ such that $xRy$ according to some predetermined distribution over the solution set $\{y \in \Sigma^* : xRy\}$ if one exists.
4. **Counting**: Compute $|\{y \in \Sigma^* : xRy\}|$.
5. **Enumeration**: Return all $y \in \Sigma^*$ such that $xRy$.

Of these, generation and enumeration are clearly relevant to history filtering. Prior work (Schmid *et al.* [2021]; Brown *et al.* [2020]; Moravčík *et al.* [2017]; Brown and Sandholm [2019]) has generally relied on enumerative methods, i.e. filtering histories by explicitly representing the entire PBS. Generative methods for history filtering potentially have the advantage of avoiding explicit PBS representation. In the next section, we analyze the computational complexity of problem variants 1-5.

# 4 Complexity of Filtering Histories

Efficient (polynomial-time and space) algorithms for history filtering enable scalable depth-limited search in imperfect information games. In this section, we provide an FOSG instance where the construction variant of FILTER is intractable and explain when efficient enumeration is possible.

## 4.1 FNP-Completeness of Construction

Consider the following two-player FOSG, based on Functional Boolean Satisfiability (FSAT):

**Definition 2.** *(3-FSAT-GAME) For a given integer $m$, world states are encoded as $m$-variable truth assignments. Starting at initial state $w^0$, player 1 chooses an action $a^0 = (y_1, y_2, ..., y_m)$ that represents a truth assignment. This is followed by a transition to some $w = (y_1, y_2, ..., y_m)$ that encodes the same assignment. $O_{pub}(w^0, a^0, w)$ reveals no public information about the transition to $w$ except that the action and transition occurred. $w$ has joint action set $\mathcal{A}(w) := \{a\}$ and $\mathcal{T}(w, a) := w$ for all $w \in \mathcal{W}$. Rewards are arbitrary. When action $a$ is taken at time $t$, the public observation function $O_{pub}(w, a, w)$ generates, at random, a 3-CNF clause $c_t$ that is satisfied by $w$.*

Player 1 chooses an $m$-variable truth assignment, and then player 2 repeatedly takes arbitrary actions—outputting a 3-CNF clause satisfied by the assignment as a public observation each time. The truth assignment is unknown to player 2, so solving a 3-CNF Boolean formula is necessary to construct a history consistent with the public observations. Additionally, a sequence of observations equivalent to any satisfiable 3-CNF formula over $m$ variables can be generated by playing this game.

**Theorem 1.** *There exists a joint policy $\pi$ for which the construction problem associated with* `FILTER(3-FSAT-GAME, `$\pi$`)` *is FNP-complete.*

Theorem 1 implies that unless $P = NP$, computing even a single history corresponding to a given public state in the `3-FSAT-GAME` is intractable in the worst case. It follows that the same applies to more complex computations such as generation and enumeration. However, there are several examples where these computations have been successfully performed in practical examples of games such as poker—we discuss where efficient enumeration is feasible next.

## 4.2 Efficient Enumeration in Games with Sparse Public States

Prior work has often limited application domains to games where the public state is trivially enumerated and beliefs can be represented explicitly in physical memory; the efficiency of these algorithms depends on a structural property of the game tree that we call *sparsity*.

**Definition 3.** *The public tree $\mathcal{S}$ of an FOSG $G$ is **sparse** if and only if all public states $S = (o^1, o^2, ..., o^t)$ in $\mathcal{S}$ satisfy $|\mathcal{H}_S| \leq p(t)$ for some polynomial $p$. Public trees that do not satisfy this property are **dense**.*

Public states in games with sparse public trees can be enumerated in polynomial time using a simple breadth-first search that makes at most $|\mathcal{A} \times \mathcal{W}|p(k)$ calls to the observation function at depth $k$ (see proof of Theorem 2 in the appendix for more details).

**Theorem 2.** *For any FOSG $G$ with finite $\mathcal{W}, \mathcal{A}$, the enumeration problem associated with* `FILTER`$(G, \pi)$ *can be solved in polynomial time if and only if $G$'s public tree is sparse.*

As an example, consider the following variants of two-player poker. In Texas Hold'em, there is a 52-card deck and two cards are dealt to each player; public states in this game are of constant size. With $n$ cards in the deck where each player is dealt 2 cards, the number of histories per public state is polynomial in $n$. Both of these games have sparse public trees. However, with $n$ cards in the deck and $k$ cards dealt to each player, the number of histories is exponential in $k$, so the public tree is dense.

Sparsity itself does not guarantee that enumeration is feasible on modern hardware. For instance, physical memory and time constraints still prevent enumeration in large trick-taking card games with constant-sized public states such as Skat and Hearts. In the next section, we propose an algorithm for history generation in these games and validate it empirically.

## 5 MCMC History Generation in Trick-Taking Card Games

Trick-taking card games (TTCGs) like Contract Bridge, Skat, and Hearts are played by millions of people worldwide and have historically been of significant interest to the AI community. However, superhuman-level computer play in these games has yet to be achieved—in part because of their large public belief states. Here, we devise a Gibbs sampler (Geman and Geman [1984]) for history generation in TTCGs that treats histories as states in a Markov chain and uses local computations to generate histories without explicit belief representation.

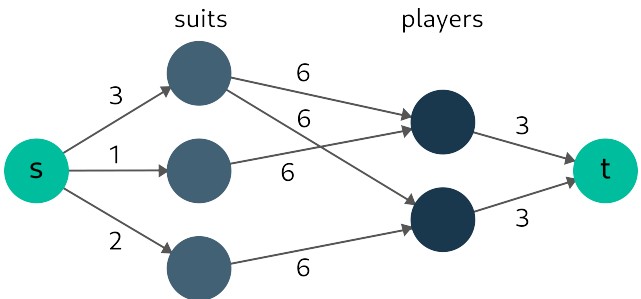

Figure 3: A flow network constructed from a history in a TTCG with three suits and two players. The suits have three, one, and two unknown cards remaining, respectively, and each player must be dealt three more cards. The first player is void in the third suit, and the second player is void in the second suit.

## 5.1 Challenges of Filtering in TTCGs

Depending on the size of the deck and the number of cards dealt to each player, public states in TTCGs may be large early in the game: up to $n!/(n-k)!$ for decks with $n$ cards and $k$ cards dealt in total. They shrink as information is revealed because players must be dealt the cards they play, and *void suits* (when a player reveals they cannot have a certain suit through play) imply certain cards cannot be dealt to certain players. These observations lead to two possible constraints on a card being dealt to a player: either it must have been dealt to a player, or it cannot have been dealt to a player. Both can be checked efficiently through a single pass over the observation sequence.

## 5.2 History Construction in TTCGs

The solution to history construction for some $S \in \mathcal{S}$ is a history $h \in \mathcal{H}_S$. This history can serve as an initial state for the Markov process we have designed to solve generation in TTCGs. As we now describe, construction can be solved in polynomial time using an algorithm for integer maximum flow such as Edmonds-Karp (Edmonds and Karp [1972]) along with some simple pre- and post-processing steps.

Given a history $h$ with the subsequence of private actions that represent the deal, $\sigma \sqsubseteq h$, we create a flow network that captures the constraints of the cards in $\sigma$. Cards revealed through play must be dealt to the player that played them, so we can ignore them when solving for deals that satisfy the other constraint: where a player cannot have any more cards in a suit. The source vertex is connected to $k$ *suit* vertices via directed edges with a capacity that corresponds to the number of unknown cards remaining in that suit. Each suit vertex is connected to a *player* vertex if it is possible for that player to hold cards of that suit in their hand. The edges connecting the player and suit vertices have capacity equal to the number of total unknown cards remaining in all suits. Finally, the player vertices are connected to the sink via edges that correspond to the number of cards remaining in that player's hand. See Figure 3 for an example.

**Lemma 2.** *For TTCG $G$ and joint policy $\pi$ with full support, FILTER$(G, \pi)$ can be solved in polynomial-time using a maximum flow computation.*

Applying a maximum flow algorithm based on augmenting paths will construct integral flows that represent assignments of the number of unknown cards in each suit dealt to each player—which we call a **suit length assignment**. We can select one of possibly many histories that satisfy the suit length assignment (and the cards that have been explicitly revealed through play) and construct a history from $\mathcal{H}_S$.

## 5.3 TTCG Gibbs Sampler

The TTCG Gibbs sampler is Markov Chain Monte Carlo method (see Häggström and others [2002] for an overview) for generative history filtering in TTCGs. It is based on two concepts: Markov chain states are histories from $\mathcal{H}_S$, and transitions use unnormalized reach probabilities and involve local modifications to the suit length assignment of the current state.

---

**Algorithm 1:** RingSwap

---

**input** : $S$ — public state, $\sigma$ — deal consistent with $S$

**output** : $\Omega_\sigma$ — set of neighbors of $\sigma$

1 **let** $A_{n \times m}$ be the suit assignment matrix for $\sigma$

2 $\Omega_\sigma \leftarrow \{\}$

3 **for** *Row $i$ in rows(A)* **do**

4     **for** *Columns $j, k, j \neq k$ in cols(A)* **do**

5         **if** void$(S, i, j)$ or void$(S, i, k)$ or $a_{i,k} = 0$, **continue**

6         $C \leftarrow A$

7         $c_{i,j} \leftarrow c_{i,j} + 1; c_{i,k} \leftarrow c_{i,k} - 1$

8         $\Omega_\sigma \leftarrow \Omega_\sigma \bigcup \text{BFS}(C, n)$ to find all ways to make $C$ a valid suit length assignment

9     **end**

10 **end**

11 **return** $\Omega_\sigma$

---

We start by describing the neighbor generation algorithm, RingSwap (Algorithm 1), which operates on suit length assignment matrices. Constraints on suit length assignments can be encoded using a matrix with row and column sums equal to the number of unknown cards in each suit and player hand. Void suits are represented as entries fixed to zero. For example, the following is a suit length assignment matrix that satisfies the max flow network in Figure 3:

$$\begin{bmatrix} 2 & 1 & 0 \\ 1 & 0 & 2 \end{bmatrix}$$

The rows sum to 3 because each player has 3 unknown cards, and the columns sum to the number of unknown cards in the corresponding suit.

With suit assignment matrix $A$ corresponding to $\sigma$, RingSwap repeats the following for all players $i$. For every pair of non-void suits $j$ and $k$, perform a *swap* by adding a card to $A_{i,j}$ and removing one from $A_{i,k}$. The column sums of the matrix are now incorrect ($j$ has too many cards and $k$ has one too few), and must be corrected via a sequence of swaps in other rows. All sequences of swaps of length $< n$ that lead to valid suit length assignments are then computed via BFS and a valid assignment is selected proportionally to the number of histories it corresponds to.

We can now describe the *TTCG Gibbs sampler*. For public state $S$ at time $t$, given $X_t = h$, deal $\sigma \sqsubseteq h$, and a joint policy $\pi$ with full support at all infostates, consider the following Markov chain:

At time $t + 1$:

1. Compute $\Omega_\sigma$, the set of all neighbors of $\sigma$ using the procedure RingSwap$(S, \sigma)$

2. Sample $\sigma'$ uniformly from $\Omega_\sigma$

3. Compute $\Omega_{\sigma'}$ and $h'$ such that $\sigma' \sqsubseteq h'$ by replacing $\sigma$ with $\sigma'$ in $h$ to form $h'$

4. Let $z = \min\{1, \frac{\bar{P}^\pi(h')|\Omega_\sigma|}{\bar{P}^\pi(h)|\Omega_{\sigma'}|}\}$

5. With probability $z$, $X_{t+1} = h'$, otherwise $X_{t+1} = h$

State transitions are done according to the Metropolis-Hastings algorithm (Metropolis *et al.* [1953]; Hastings [1970])— with unnormalized reach probabilities $\bar{P}^\pi$ as $\mu^*$ and uniform selection over the neighboring states. All computations are local to the current history at time $t$, and take at most polynomial time in the history length. The following theoretical details state that the chain is suitable for unbiased history generation.

**Theorem 3.** *The TTCG Gibbs sampler is aperiodic and irreducible.*

Theorem 3 implies that the TTCG Gibbs Sampler converges to some stationary distribution; the following theorem ensures that its stationary distribution is the desired $P^\pi$.

**Theorem 4.** *The stationary distribution of the TTCG Gibbs sampler with input $\pi$ is $P^\pi$.*

Given an initial history obtained by solving the construction problem for the game and a policy with full support at all infostates, the TTCG Gibbs sampler correctly generates histories from $P^\pi(\cdot|S)$ in

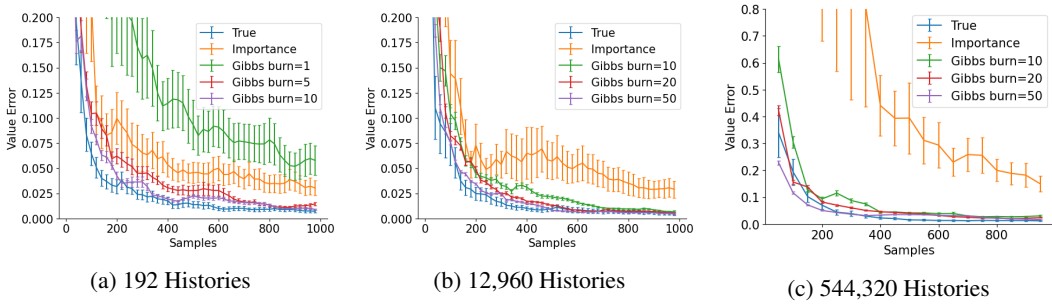

(a) 192 Histories      (b) 12,960 Histories      (c) 544,320 Histories

Figure 4: Value estimation error of TTCG Gibbs Sampler with specified burn-in and baselines on PBS of various sizes. Error bars show one standard error of the mean over 100 runs. See the appendix for full game parameters.

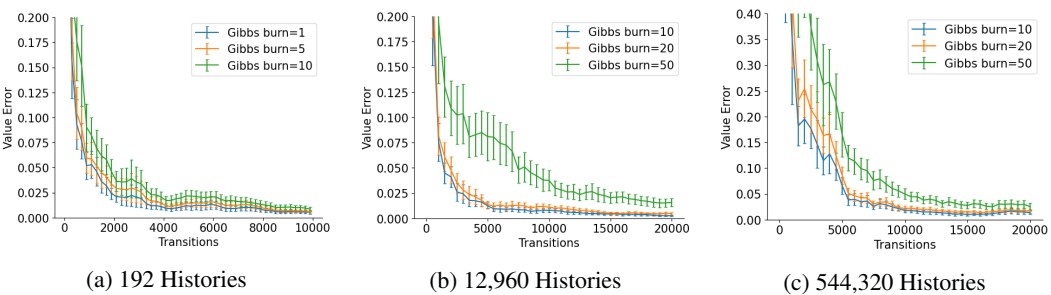

(a) 192 Histories      (b) 12,960 Histories      (c) 544,320 Histories

Figure 5: The effect of the number of samples burned on the value estimation error of the TTCG Gibbs Sampler. Burning fewer samples can result in a lower value error after an equal number of transitions. Error bars show one standard error of the mean over 100 runs.

the limit. The next section validates the efficiency and approximation quality of the TTCG Gibbs sampler empirically in the domain of Oh Hell (Parlett [2008]); we leave the theoretical analysis of its mixing time to future work.

### 5.4 Experiments

The TTCG Gibbs sampler we just described runs in polynomial-time with respect to history length and removes dependencies on explicit belief representation. Thus, the algorithm's scalability depends on the mixing time of the underlying Markov chain. The following empirical results suggest that the chains mix rapidly in this domain.

We evaluate the TTCG Gibbs sampler using a value estimation task in the trick-taking card game Oh Hell. Oh Hell is an $N$-player game with $n$ cards in the deck, where each player is dealt between 1 and $\lfloor (n-1)/N \rfloor$ cards. We control the size of randomly generated $\mathcal{H}_S$—starting small and then scaling to $\mathcal{H}_S$ several orders of magnitude larger—by varying the number of cards in the deck and the number of cards dealt to players. The task is to estimate the expected value of $S$ under $\pi$, $V^\pi(S) = \sum_{h \in S} P^\pi(h) V^\pi(h)$, where the value of a history $h$, $V^\pi(h) = \sum_{z \in \mathcal{Z}: h \sqsubseteq z} P^\pi(h, z) u(z)$ is the expected utility of the terminal histories in $\mathcal{Z}$ reachable from $h$. The initial state is chosen uniformly from $\mathcal{H}_S$, and ranges are constructed using policies learned via independent Q-learning (Claus and Boutilier [1998])—hyperparameters and other details can be found in the supplementary material. Strong performance in this task does not necessarily imply improved search performance; instead, it demonstrates the sampler's ability to generate histories from the correct public belief state.

Figure 4 shows the value error curves of the TTCG Gibbs sampler with a specified burn-in compared to two Monte Carlo baselines. `True` samples from the exact joint range, and `Importance` performs (biased) importance sampling with a uniform proposal and unnormalized reach probabilities. The latter estimates the PBS value by correcting the weight of samples drawn uniformly from $\mathcal{H}_S$. In practice—especially at larger scales than these experiments—importance sampling may not be feasible without a generation algorithm that approximates a known proposal distribution in polynomial time. We see that the TTCG Gibbs sampler outperforms importance sampling and closely

approximates sampling from the joint range in all cases while using a burn-in orders of magnitude smaller than the size of the public belief states.

Figure 5 shows the effect of burning samples on value estimation error. On one hand, our results show that burning fewer samples can result in a better value estimate with fewer state transitions. This is unsurprising because larger sample sizes generally lead to better estimates. However, evaluating a sample could be several orders of magnitude more costly than performing a state transition in the Markov chain in practice. Comparing Figures 5 and 4 shows that, if the estimation task is constrained by time or computational resources, fewer high-quality samples may produce a better estimate. The appropriate burn-in depends both on the task and the available resources.

Our experiments show that the TTCG Gibbs Sampler leads to increased scalability over the enumerative approach in this value estimation task. With a small burn-in of 20 state transitions, a good approximation of the PBS value is achieved after only 400 samples and therefore only 8,000 total state transitions—compared to the enumerative approach which must construct and calculate the reach probability for 544,320 histories. State transitions are computed locally to the current state (history) and do not require knowledge of the rest of the PBS, so memory requirements are dramatically reduced. These effects increase along with public state size; in Oh Hell with a 52-card deck, public states can have over $10^{61}$ histories.

## 6   Discussion and Conclusions

In this paper, we analyzed the computational complexity of history filtering for subgame decomposition and depth-limited search in imperfect information games. Although even the simplest form of the computation is likely intractable in general, we have shown that depth-limited search remains a viable option in certain classes of imperfect information games. Efficient enumeration is achievable in games with polynomially-sized public states; many application domains from prior work seem to have this property. However, generative methods for history filtering may not require explicit belief representation, and are therefore more scalable than enumeration. To this end, we have introduced a novel generation algorithm for trick-taking card games for asymptotically correct and efficient history filtering that fits seamlessly with Monte Carlo-style search algorithms.

Our TTCG Gibbs sampler needs a method that constructs a valid history from the public state but is otherwise flexible. It does not depend on knowing the size of the public state and does not require normalized reach probabilities to produce samples with the correct probability. Unlike the enumerative approach which front-loads all of its computation, our algorithm applies to the setting where the game-playing algorithm must return an action within a predefined time budget. It is also easy to parallelize; multiple samples can be generated simultaneously by starting the process multiple times from the same initial state and running each for the desired burn-in time. We demonstrate its effectiveness empirically—though future work should prove rapid mixing analytically.

## Acknowledgements

This work was funded by the Canada CIFAR AI Chairs Program. We acknowledge the support of the Natural Sciences and Engineering Research Council of Canada (NSERC), Prairies DRI, and the Digital Research Alliance of Canada.

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
