# OpenReview forum: "History Filtering in Imperfect Information Games: Algorithms and Complexity"
_NeurIPS.cc/2023/Conference — NeurIPS 2023 poster_

### Official Review · Reviewer_8L8P · 2023-06-18

**Soundness:** 3 good
**Presentation:** 1 poor
**Contribution:** 3 good
**Rating:** 7
**Confidence:** 4

**Summary:**

The submission considers the problem of approximating public belief states. The submission's main contributions are:
- Showing that certain fundamental computational problems related to public belief states are generally FNP-complete.
- Defining a subclass of games and showing that enumerating public belief states for these games can be done in polynomial time.
- Introducing an MCMC algorithm for the computation of public belief states in trick-taking card games.

**Strengths:**

I think that studying scalable mechanisms for approximating public belief states is an important direction, so I am partial to the direction of the submission. Each of (what I view as) the submission's main contributions is novel. Both Theorem 1 and Theorem 2 are nice results. The MCMC algorithm is also a good contribution.

---

I am disclosing here that I did not read the proofs of the theorems or check the details of the experimental setup.

**Weaknesses:**

My criticisms regarding the submission mostly hinge on presentation.

I think the submission does a poor job with respect to existing work in two respects.

**First, it does not motivate the importance of public belief states as well as it could.** The only motivating use cases provided in the introduction are DeepStack and Pluribus. There are so many more great works and lines of research that rely on public belief states:

> Work in control literature: Decentralized Stochastic Control with Partial History Sharing: A Common Information Approach (Nayyar et al., 2013)

The paper that introduced public belief states! There is a large body of work in decentralized stochastic control literature using public belief states derived from Nayyar et al.'s work.

> Work in Dec-POMDP literature: Optimally solving Dec-POMDPs as continuous-state MDPs (Dibangoye et al., 2013); Sufficient Plan-Time Statistics for Decentralized POMDPs (Oliehoek 2013)

Independently discovered public belief state-like objects for Dec-POMDPs. There are large bodies of on how Dec-POMDPs can be solved using HSVI from this perspective.

> Search algorithms for common-payoff games: Improving Policies via Search in Cooperative Partially Observable Games (Lerer et al., 2020); Scalable Online Planning via Reinforcement Learning Fine-Tuning (Fickinger et al. 2021); A Fine-Tuning Approach to Belief State Modeling (Sokota et al., 2022)

The best performing Hanabi AIs rely on search techniques that use public belief states!

> Other poker AIs: Superhuman AI for heads-up no-limit poker: Libratus beats top professionals (Brown and Sandholm, 2017); Combining Deep Reinforcement Learning and Search for Imperfect-Information Games (Brown et al., 2020); Player of Games (Schmid et al., 2021)

I'm sure the authors are aware of these. But it's still valuable to list them to emphasize to the reader the ubiquity of the role that public belief states have played in successful poker AI.

> Other techniques for zero-sum games: HSVI can solve zero-sum Partially Observable Stochastic Games (Delage et al., 2022)
 Abstracting Imperfect Information Away from Two-Player Zero-Sum Games (Sokota et al., 2023)

These works give new techniques for solving and doing search that don't require gadget games.

> Work for adversarial team games: A Marriage between Adversarial Team Games and 2-player Games: Enabling Abstractions, No-regret Learning, and Subgame Solving (Carminati et al., 2022); Team belief dag form: A concise representation for team-correlated game-theoretic decision making (Zhang et al., 2022)

These works show how adversarial team games can be solved using public belief states.

---

It's to the benefit of everyone for the submission to motivate itself more completely:
- Readers without extensive knowledge of literature will be left with a greater sense of the importance of public belief states. As is, the submission risks them being left with the impression that they have only been studied by a niche community mostly focused on poker.
- Authors of these works who normally interact with mostly or entirely disjoint bodies of literature from that of the imperfect information game community (such as those who work on Dec-POMDPs or decentralized stochastic control) are more likely to be made aware of the submission and in turn promote it among their own communities.
- The submission is made stronger because it is made clear that it impacts a much broader body of literature than the current version reads as impacting.

**Second, the submission appears to be entirely unaware of the single most related work.** The submission claims that applications of "theoretically-sound depth-limited search in imperfect information games ... have been limited to games in which the relevant information is
small enough to be enumerated." **This is not true!** The main contribution of *A Fine-Tuning Approach to Belief State Modeling (Sokota et al., 2022)* introduces a technique (called belief fine-tuning) for approximating public belief states in settings where the relevant information cannot be enumerated. They show that belief fine-tuning facilitates strong performance in such settings (specifically, variants of Hanabi).

I encourage the authors to compare/contrast it to their own work in a related work section. It would also be interesting to see a direct comparison with the TTCG Gibbs sampler in Oh Hell, but it would be a lot of work for a rebuttal period, so it is not a necessity.

### re: theory

I think the way that the theoretical results are summarized could be improved. In section 3.2, the submission states that the subsequent section examines problem variants 1-5. However, at a surface level read it is difficult to discern whether this is the case. The subsequent section states that the construction problem is FNP (and therefore more complex computations are, also). But what about existence? I think explicitly referring back to problem variants by number, where possible, would help to improve clarity.

### re: experiments

I would be careful in extrapolating too much from the results of the experiments. The policy class that the submission examines is not resemblant of policies that humans use or that RL models learn. Furthermore, value error does not necessarily translate well to search performance. (This is not to say that the experiments are not appreciated, just that the submission should caution the reader of these limitations.)

### Specific comments I made while I was reading (some redundancy with above)

>  Search in imperfect information games requires subgame decomposition—a process in which a subgame is computed from public information and player beliefs.

This isn't true. It's perfectly possible to perform search without subgame decomposition (for example, IS-MCTS). What you really ought to argue here is that the approaches that have been most successful/have the strongest guarantees require subgame decomposition.

> Applications have therefore been limited to games in which the relevant information is small enough to be enumerated. Scaling to larger, more complicated imperfect information games requires a deeper theoretical understanding of history filtering.

It is not true that applications have been limited to games in which relevant information is small enough to be enumerated. See: *A Fine-Tuning Approach to Belief State Modeling* (ICLR 2022). Thus, it is also untrue that scaling to such games requires a deeper theoretical understanding of history filtering than existed prior to the submission.

> Of these, generation and enumeration are clearly relevant to history filtering. Prior work (Schmid et al. [2021]; Brown et al. [2020]; Moravcík ˇ et al. [2017]; Brown and Sandholm [2019]) has generally relied on enumerative methods, i.e. filtering histories by explicitly representing the entire PBS. Generative methods for history filtering potentially have the advantage of avoiding explicit PBS representation.

Generative approaches in combination with public belief states in both of the works below:

*Scalable Online Planning via Reinforcement Learning Fine-Tuning* (NeurIPS 2021)

*A Fine-Tuning Approach to Belief State Modeling* (ICLR 2022)

The reason these works avoided explicit PBS representation was actually orthogonal to PBS representations. For the NeurIPS 2021 paper, it was because of the high computational expense of doing search for every information state supported by the PBS. For the ICLR 2022 paper, it was to scale to settings in which the support of the PBS is too large to enumerate.

> Prior work has typically limited applications to games where the public state is trivially enumerated and beliefs can be represented explicitly in physical memory

Again, this is untrue -- see:

*A Fine-Tuning Approach to Belief State Modeling* (ICLR 2022)

**Questions:**

>  Think of the things where a response from the author can change your opinion

As articulated above, my main concerns are regarding the presentation. (Despite the disproportionate amount of text in the weaknesses box, I liked the paper : ) !) A revised copy of the submission that addresses the presentation criticisms would change my opinion. If this is not allowed during the review period, a provision of detailed text that the authors commit to including in the paper may change my opinion.

**Limitations:**

I think the place where the submission is in most danger of not having acknowledged its weaknesses is in the experiments section. The submission makes claims such as "We demonstrate its effectiveness empirically" (it referring to the MCMC algorithm).

However, as discussed in the weaknesses section, there are some significant limitations:
- The randomly generated policies that the submission uses do not possess the kind of structure that policies that humans or neural network-based policies have -- empirical results won't necessarily extrapolate.
- The relationship between value error and search performance is hard to quantify. Having low or high value error isn't always a good indicator of downstream search performance.

---

> ### Author Rebuttal · Authors · 2023-08-08
>
> The reviewer's suggestions concerning related work and motivation have provided us with the means to significantly improve the presentation of the paper. This will be accomplished by modifying the abstract and introduction, replacing Sections 2.2 and 2.3 with text that motivates public belief states in a broader sense, and also text discussing the main differences between our approach and belief fine-tuning.
>
> For the abstract and introduction, the impact of our submission does not necessarily depend on the untrue claims that 1) subgame decomposition is necessary for search in imperfect information and 2) generative methods are completely novel. For 1), as stated by the reviewer, algorithms with strong guarantees that tend to perform best require subgame decomposition. For 2), our generative approach is novel because it works in public states with support too large to enumerate and is unbiased without having to generate
> and reject inconsistent histories. Ideally, we would also compare empirically to belief fine-tuning to show that this makes a difference for search, but we have not yet done so.
>
> Moving the necessary notation to clearly explain Sections 3-5 that is currently in Section 2.2 and Section 2.3 to Section 2.1, while replacing those sections with more motivation related to HMMs, POMDPs, and Dec-POMDPs - as well as the recent work on zero-sum POSGs. FOSGs are generalizations of HMMs and POMDPs, so citing the mentioned papers is sufficient for those bodies of work. Likewise, the missing work on Poker and Hanabi AI will be mentioned for completeness.
>
> The most important improvement our submission can make is comparing our method with BFT. BFT takes a pre-trained model as input, whereas our method is parameter-free, given the policy. One of the main advantages of BFT is that it doesn't require training the sequential generative model as a function of the policy. This is vital for the FOSG case that we focus on, and our method achieves the same goal by avoiding training that model altogether. Our approach instead requires efficient domain-specific algorithms for the construction problem and neighbor generation in the Markov chain that provably leads to irreducibility. This could be viewed as a shortcoming compared to BFT, but in domains where our approach applies, it is unbiased - unlike BFT. Another potential advantage of our approach is that it always produces valid histories according to the game rules. In some domains other than Hanabi, it is unclear how to structure the pre-trained model such that its output is always valid. Otherwise, the dynamics model needed for fine-tuning may be undefined. A concrete example of where this could happen is any trick-taking card game, where a naively structured belief state model could assign a player a card that they could not possibly hold according to the game's rules. As we show formally earlier in the paper, producing histories that correspond to observation sequences is a hard problem in itself. Our algorithm comes with clear requirements that guarantee it will apply correctly.
>
> We will also caution the reader of the limitations of the experiments and emphasize that they are primarily intended to demonstrate the efficiency of the approach (in terms of mixing time) while still maintaining reasonable accuracy (by outperforming importance sampling). Though our experiments demonstrate that our approach is effective in the sense that we can use it to approximate PBS values in the example domain, it should be clear to the reader that this does not imply that the method will necessarily lead to better search performance.
>
> In any case, the detail in this review is exceptionally helpful for the improvement of our paper.

---

> > ### Comment · Reviewer_8L8P · 2023-08-11
> > **Response to Rebuttal**
> >
> > Based on the author response, I am satisfied that my concerns will be addressed in the camera-ready version of the work and am raising my score accordingly.

---

### Official Review · Reviewer_r9WW · 2023-06-23

**Soundness:** 4 excellent
**Presentation:** 4 excellent
**Contribution:** 3 good
**Rating:** 6
**Confidence:** 4

**Summary:**

This paper studies the algorithmic complexity of history enumeration and generation in imperfect information games. The authors define these problems with a formal model of factored observation stochastic games (FOSGs). They proved computational complexity results, and empirically demonstrated the effectiveness of an MCMC algorithm for generating history in the domain of trick-taking card games.


**Strengths:**

The problem studied in this paper is interesting, and the authors take an original perspective. In particular, they observe the centrality of history enumeration and generation in state-of-the-art algorithms for depth-limited game solving, and provide rigorous definitions of the associated computational problems.

The work appears technically sound: both the theoretical complexity analysis and experimental study of MCMC methods for history generation.

The presentation is clear, both in prose and mathematical description.


**Weaknesses:**

The major theoretical conclusion is that enumerating history is algorithmically hard in general, but is feasible for games that are sparse in a well-defined sense. These results are not surprising, and the authors do not pull out any deeper insights from their demonstration.

The two parts of the paper—fundamental complexity analysis and application to card games—seem quite disjoint from each other.



**Questions:**

The general hardness of history enumeration for FOSGs is to be expected, as the analogous problems in a single-agent context are already understood to be hard. For example, in POMDP solving one of the challenges is to approximate belief states. Could results and methods from the POMDP literature directly apply to the FOSG setting? For example, what would any of these works have to say about the multiagent version of the problem:
[1] Nonapproximability Results for Partially Observable Markov Decision Processes, Lusena et.al.
[2] What Makes Some POMDP Problems Easy to Approximate? Tsu et.al.

Additional detailed questions:

As noted above, the MCMC empirical part reads as quite a bit detached from the theoretical part of the paper. Is the point really to exercise the theory, or is it a separate contribution to develop techniques for trick-taking card games?
Is the MCMC algorithm specific to this  domain or can it be generalized to a broader class of imperfect information games? Can you come up with a more general characterization of world-state structures for which this will be applicable and effective?





**Limitations:**

Enumerating history in imperfect information games itself is indeed an interesting combinatorial problem that is worthwhile studying. However, will this “enumeration” approach be plausible for more complex games? Computing exact beliefs seems tractable for Poker, but what about games like Stratego? In fact, it is unclear the relationship between an accurate belief state representation and the strength of a depth-limited search algorithm. The IS-MCTS algorithm the authors cite actually used a uniform distribution over possible world states to approximate a belief (instead of the exact belief written by the authors), which also renders fairly good empirical results. More efficient ways could be to use particle filtering [1], or deep generative models [2, 3].

[1] Monte-Carlo Planning in Large POMDPs, Silver & Veness
[2] Generalized Beliefs for Cooperative AI, Muglich et. al.
[3] Combining Tree-Search, Generative Models, and Nash Bargaining Concepts in Game-Theoretic Reinforcement Learning, Li et.al.

---

> ### Author Rebuttal · Authors · 2023-08-09
>
> The non-approximability results mentioned by the reviewer are of interest to our area of work in general, but concern the hardness of computing or approximating solutions to POMDPs (in the form of optimal policies) rather than finding the sequence of unobservable states corresponding to some input sequence of observations. Those results also apply to the FOSG setting, but not in the context of history filtering.
>
> We note the clarity issues raised by the reviewer, and want to emphasize that the sparsity result is meant to explain when enumeration is a viable option. Since it is not always the case, we introduce an approximate, but unbiased, generative method, similar to particle filtering, in the second part of the paper. Our MCMC method is intended for domains where computing exact beliefs is intractable. Games like Stratego are an intended use case, and our experiments are evidence that it can be a viable approach in domains where we can develop efficient algorithms for construction and neighbor generation in the Markov chain.

---

> > ### Comment · Reviewer_r9WW · 2023-08-16
> >
> > I have read and appreciate the author response.

---

### Official Review · Reviewer_KG55 · 2023-07-01

**Soundness:** 2 fair
**Presentation:** 4 excellent
**Contribution:** 3 good
**Rating:** 3
**Confidence:** 4

**Summary:**

The paper presents theoretical analysis on the hardness of history generation in public belief states, a concept used in imperfect information games (IIGs). Then for a game where enumeration is prohibitively expensive (trick-taking card game Oh hell), they devise a specialized Gibbs sampler to generate histories from the public belief state. The sampler uses a polynomial time algorithm based on solving integer maximum flow to generate history candidates from the corresponding public state. The candidates that are then filtered to correct their distribution, i.e. to find the public belief state. In experiments, they compare the Gibbs sampler with baselines.



**Strengths:**

1. Clearly written and well motivated paper. It can serve as a reference for claiming whether a game is "hard" or "easy" to generate histories from a PBS.
2. Introduces a practical generative algorithm without explicit range representation for estimating PBS value in game Oh Hell.
3. Shows that Gibbs sampler can be indeed used for their setting.

**Weaknesses:**


1. A weakness of the paper is misrepresentation of prior work, specifically the claim that tractability has been ignored and that only games like poker were considered.

   Consider the quote from the cited paper [Richards and Amir, 2012]:

   > Identifying legal moves involves testing the satisfiability of arbitrary Boolean expressions and is therefore theoretically exponential in the size of the game description.

   This paper involved history generation in the game of Racko, Battleship and GOPS.

   There is a missing citation to [1], which also claims the problem is difficult:
   > Due to exponential branching of the opponent’s private information, a large portion, or even all, of the tracked states may suddenly become incompatible with a received observation.

   The paper involved history generation in the game Stratego, Phantom Tic-Tac-Toe, Goofspiel.

   I concede there were no _formal_ definitions on tractability. But the claim seems too strong, as previous quotes show there was awareness on the difficulty of the problem.

2. Def 3 - I believe the definition of a sparse public state is not well formulated, for what Thm 2 attempts to state. For any game, we can take polynomial $p$ of high enough degree such that we can then claim any $G$ is always sparse.  Then, according to Thm 2., solving FILTER(G,$\pi$) is always polynomial time. But this is inconsistent with Thm 1. I believe this could be resolved by adding a restriction on the degree of the polynomial $p$.

3. The paper introduces the concept of sparse public states to show the enumuration for FILTER problem can be done in polynomial time. However, authors do not take advantage of this separation of "easy" and "hard" problems. I would expect authors to use the bound to separate some games based on their hardness and an experiment that shows performance of a general algorithm on the two classes and compare that to a baseline. Instead, they found a game that is "hard" according to the bound, and devised a domain-specific algorithm that can approximate PBSs in that game.

4. For the PBS value error experiment I am missing a motivation for why this is a relevant metric. As far as I know, existing algorithms in prior work use rather infostate-value functions for approximating strategies and PBS values are not descriptive enough for the use in IIGs. Can you add a motivation, perhaps by a reference to a paper  where this would be relevant?

[1] Learning to guess opponent’s information in large partially observable games. D Seitz*, N Milyukov, V Lisy.


**Questions:**

1. In experiments, how is the value of a history $V^\pi(h)$ computed? I assume this is not part of the compared algorithms and it is exactly computed?
2. Can the method be trivially applied to other trick-taking card games like the mentioned Contract Bridge, Skat, and Hearts, or it is applicable only to Oh Hell? Are there trick-taking card games that have sparse PBSs (i.e. are "easy")? If yes, it would make the paper a lot stronger if your algorithm is shown to be more general for a class of trick taking games, and that it can generate PBS in both "easy" and "hard" games.

**Limitations:**

Yes.

---

> ### Author Rebuttal · Authors · 2023-08-09
>
> Our response to Reviewer 8l8p describes how we will address potential misrepresentations of prior work.
>
> It's true that for any fixed, finite game, we can choose some polynomial with arbitrarily high degree to bound the size of PBS support. However, complexity arguments, such as sparsity, only apply when we consider games or family of games that scale in size.
>
> The clarity problems in our work has lead to confusion about sparsity, the motivation for generative algorithms and our MCMC approach, and our experiments. The concept of sparsity is meant to explain when enumeration may be a viable approach in certain families of games because there PBS sizes only grow polynomially. It provides a rule of thumb to understand where important algorithms from prior work are applicable as-is.
>
> Oh Hell is an example game where enumeration is not viable at scale, but our new MCMC approach can approximate beliefs. The domain is not chosen to be hard according to our bound, but rather to represent basic trick-taking card games, which have long been of significant interest to the community. Our experiments are in small versions of Oh Hell, and still enumerate the public state to provide an upper bound on what any sample-based generative method could hope to achieve. We also compare to an importance sampling baseline.
>
> The rules of Contract Bridge introduce new constraints (other than 1. player must have card $c$ and 2. player cannot have suit $s$), and thus the game likely requires a different neighbor generation algorithm (and potentially a different construction algorithm as well). However, we see no reason why the algorithm would not apply to Skat and Hearts - where the constraints are the same as Oh Hell.
>
> ReBel, introduced in Combining deep learning and search for imperfect information games (Brown et al., 2020), demonstrates that infostate values can be recovered from the expected values of public states that contain them.
>
> $V^\pi(h)$ is indeed computed exactly in our experiments.

---

### Official Review · Reviewer_684B · 2023-07-03

**Soundness:** 4 excellent
**Presentation:** 2 fair
**Contribution:** 3 good
**Rating:** 5
**Confidence:** 5

**Summary:**

The paper concerns history-filtering method to estimate values of imperfect-information subgames. Such a sub-task proved to be a crucial component of previous depth-limited solving algorithm developed in the related literature.

The contribution of the paper is twofold:
- on one hand, the paper analyses the complexity of such a task from a theoretical perspective, proving its FNP-completeness in the general case, and proving that a specific condition is sufficient to make its complexity polynomial. Such a complexity is satisfied by many games customarily addressed in the literature.
- on the other hand, the paper proposes an algorithm that performs history sampling for value estimation in trick-taking games, and experimentally evaluates the performance-quality tradeoff of such an approximation.

**Strengths:**

- Novel approach for value estimation in depth-limited computation
- Technically sound and original paper
- Possibly significant direction for future research
- Clear, intuitive and on-point explanation of the background and the relations with previous literature

**Weaknesses:**

- The structure of the paper is unclear and bipartite. In particular:
	- the first half of the paper (up to section 4) takes a general view and provides general theoretical complexity results regarding any possible history-filtering-related technique
	- the second half of the paper provides a specific technique for a very specific type of game; moreover, the design of this technique is not informed by any specific actionable insight coming from the general theory developed in the previous sections
	This bipartite structure has negative effects in highlighting the crucial properties of the proposed history-filtering framework. En fact, by reading the theoretical section one gets the idea that a crucial property for the applicability of the framework is that a sparse public tree is needed. However, I argue that the proposed technique is actually much more depending on the possibility of locally sampling $\bar P^\pi(h)$ (which is crucial for effectively guaranteeing an efficient sampling), while I don't see the possibly exponential number of histories in a public state to be an impossible obstacle (and actually it may be an opportunity given the scalability of the technique).
	I think this partition inside the paper critically worsens clarity when reading
- The gain of the overall technique are not evaluated. While from the experiments it emerges that at least in trick-taking games Gibbs sampling is a useful technique with a good tradeoff in terms of performance and memory consumption, it is not clear whether sampling is worth in this context. How do the performance of subgame solving techniques worsen when the cutoff values are estimated in this way? (I.e. how bad is having a value error of 0.01? Is it possible that subgame solving algorithms are so "delicate" that sampling does not offer good enough approximation?)
- the main gain from using the technique proposed by the paper becomes evident only at the experimental section (Lines 329-335): given that $\bar P^\pi(h)$ can be easily recomputed locally when needed given the current history $h$, one can avoid to keep in memory the whole vector of $\bar P$. This should emerge more clearly in the previous sections (Final paragraph of Section 1, introduction to Section 5)

**Questions:**

I'm open to see the author's opinion on the weaknesses I've highlighted in the previous section. Other than that, I have two minor specific questions:
- Is the *"deal $\sigma$"* term employed on line 280 referring to a "suit length assignment"? If so, in which sense $\sigma \sqsubseteq h$ ?
- TTCG sampler procedure:
	- what is intended as *"replacing"* a deal with another in a history? Is it always possible for any valid suit length assignment to be replaced in a given $h$? Can this be done efficiently?
	- why is it not the case that $|\Omega_\sigma| = |\Omega_{\sigma'}|$, considering that the swap operations are commutative?

**Limitations:**

Limitation of the current analysis are properly addressed.

---

> ### Author Rebuttal · Authors · 2023-08-09
>
> Valid criticisms of the bipartite nature of the paper has led to clarity issues regarding our contributions. First, we seek to formalize the problem of history filtering and show that it is hard in general (Theorem 1). Theorem 2 then provides a condition called sparsity which describes the domains where the enumerative approach can be successful. This helps explain where some well-known algorithms from prior work (DeepStack, Player of Games, ReBel, etc.) can be applied as-is. Finally, we discuss a new approach that is necessarily more scalable and can approximate history filtering in basic trick-taking card games - a domain of significant interest to the community where enumeration is not tractable.
>
> The Reviewer claims that the fact that $\bar{P}(h)$ can only be recomputed from just $h$ emerges only in the experimental section. This is not true. However, we concede that the clarity issues discussed above could be responsible. Lines 230-233 discuss how our contribution does not depend on explicitly representing beliefs, and line 289 states that the reach probabilities can be unnormalized. We also note that the reviewer is correct that, with the scalability of our technique, the possible exponential size of the PBS is not an obstacle in domains where our algorithm applies. That is one of the main advantages of our algorithm.
>
> In terms of experiments and the gain of our overall technique, we add that the algorithm outperforms importance sampling using a small burn-in for the Markov chain. The small burn-in is evidence that the Markov Chain is rapidly mixing, which implies that our MCMC approach as a whole is efficient (i.e. samples are generated in polynomial time). We agree that future work evaluating our approach in an end-to-end algorithm that learns via solving subgames is needed.
>
> In the TTCG Gibbs Sampler procedure, a deal is a prefix history containing all "dealing" actions. One suit length assignment can correspond to many deals. As deals are prefixes of the histories, replacing is simply changing the prefixes while holding the rest of the history (the cardplay actions) constant. It is always possible unless only a single suit length assignment remains, and the procedure is efficient (see proof of Theorem 3). For neighbor generation, only a single unit is moved between two suits for any player, so $\sigma$ and $\sigma^\prime$ have different neighbor sets. Additionally, different suit length assignments may have different numbers of corresponding deals.

---

> > ### Comment · Reviewer_684B · 2023-08-18
> >
> > While the high level points expressed by the authors' rebuttal are rational and correct, I'm not seeing how the authors are going to act on the paper to avoid that other people will have the same issues I did.
> >
> > The only planned intervention on the paper I see at the time of this comment is the one regarding the previous works. This does not address any of clarity issues raised by me and other reviewers. If this is the case, my opinion will remain unchanged.

---

### Comment · Reviewer_8L8P · 2023-08-20
**Discussion**

I have read the other reviews. Some of the criticisms that I am most sympathetic to are as follows:

### Dissonance between contributions

> 684B: This bipartite structure has negative effects in highlighting the crucial properties of the proposed history-filtering framework. En fact, by reading the theoretical section one gets the idea that a crucial property for the applicability of the framework is that a sparse public tree is needed.

> KG55: The paper introduces the concept of sparse public states to show the enumuration for FILTER problem can be done in polynomial time. However, authors do not take advantage of this separation of "easy" and "hard" problems. I would expect authors to use the bound to separate some games based on their hardness and an experiment that shows performance of a general algorithm on the two classes and compare that to a baseline. Instead, they found a game that is "hard" according to the bound, and devised a domain-specific algorithm that can approximate PBSs in that game.

I agree that the flow between the contributions is a bit awkward for the reasons identified above.

### Experimental Evaluation

> 684B: The gain of the overall technique are not evaluated. While from the experiments it emerges that at least in trick-taking games Gibbs sampling is a useful technique with a good tradeoff in terms of performance and memory consumption, it is not clear whether sampling is worth in this context. How do the performance of subgame solving techniques worsen when the cutoff values are estimated in this way? (I.e. how bad is having a value error of 0.01? Is it possible that subgame solving algorithms are so "delicate" that sampling does not offer good enough approximation?)

I agree with this perspective and made a similar comment myself.

While it's worth noting that there is Hanabi literature that has successfully combined subgame solving algorithms with approximate public belief states (with value error greater than 0.01), I think this criticism remains valid.

---

I think a difference between my giving a high score and some other knowledgable reviewers giving a low score is a matter of perspective. I am primarily considering the submission on the basis of its strengths (fundamental complexity results for PBSs and a new algorithm for approximating PBSs for an interesting class of games), while some other knowledgable reviewers are focusing more on the weaknesses, like the dissonance between the contributions, the lacking experimental evaluation, etc. I don't think the perspective of the other reviewers is wrong and would not think it unreasonable of them to maintain their positions. However, I'd offer the following points for why it may be prudent to overlook these weaknesses:
1. Although public belief states are fundamental objects for imperfect information games, there is only a single existing work on approximating them at scale. Thus, it is to the benefit of the community to encourage work in this area.
2. The two weaknesses listed above are not very easy to fix:
    - Regarding the dissonance issue, breaking up the submission into two separate papers may result in two contributions that each fall short of the bar of a publishable unit.
    - Regarding the evaluation issue, adding search experiments would result in the information in the paper being more crammed and may require sacrificing clarity on other issues.

I think each of these issues can be somewhat ameliorated. Regarding the dissonance issue, the authors can give more direct commentary in Section 5 on how their complexity results from Section 4 do and don't come into play. Regarding the evaluation issue, the authors can at least use more reasonable policies to evaluate for their experiments. I.e., run some deep RL agents in self play and use those policies instead of randomly generated policies used in the submission.

I hope that, *if the authors commit to making these changes*, other reviewers may consider increasing their scores. I think it would be undesirable if the community were to miss out on the contributions of the submission entirely as a result of its weaknesses.

---

> ### Author Response · Authors · 2023-08-20
>
> We thank the Reviewer for their comments and suggestions. We will make use of them to improve the paper further.
>
> **Dissonance between contributions**
>
> We agree that our main contributions feel disjointed in the submitted version, and that fixing this completely is a difficult task. On their own, the complexity results are not sufficient to warrant publication at a suitable venue. However, they are of interest to the community for the reasons mentioned by the Reviewer.
>
> Our complexity results are foundational for the development and application of polynomial-time algorithms for history filtering. In our revision, it is vital that we clarify that our complexity analysis is what enabled us to identify a domain with *dense* public belief states, and develop a novel, scalable algorithm for filtering histories in it. Our complexity contributions provide new and more precise descriptions of the computational challenges that arise in history filtering---formalizing well-known ideas such as "large" public belief states and histories that are difficult to compute given the observation sequence. We agree with the Reviewer that these concepts are of significant interest to the community.
>
> Clarifying these points in Section 1 and the beginning of Section 5 will significantly improve the clarity and flow of the paper, and we commit to making these changes immediately. Emphasizing the importance of polynomial-time algorithms for dense public belief states will also have the side effect of improving the clarity regarding the strengths of our Gibbs sampler and Reviewer 684B's comments about $\bar{P}^\pi(h)$.
>
> **Experimental Evaluation**
>
> We agree with the Reviewer that using more reasonable policies would improve the validity of our experiments. We commit to training policies via independent Q-learning in *Oh Hell*, and use this more relevant class of policies to evaluate our method.
>
> We hope that with these changes, along with the improvements regarding related work, and the comments by the Reviewer regarding the strengths and importance of our work, other Reviewers may consider raising their scores.

---

> > ### Comment · Reviewer_684B · 2023-08-21
> >
> > I completely agree with the view expressed by 8L8P.
> >
> > The main concern in my scoring is that unclarities exist in the paper and that they are not being acted upon. I think that the suggested changes are reasonable, and the ones proposed by the authors are a fine implementation of those.
> >
> > Unfortunately it is late to personally evaluate those as we are deep in the reviewing phase. I consider my updated score in view of this extra discussion to be a borderline accept.

---

> > > ### Comment · Reviewer_8L8P · 2023-08-21
> > > **Comment to Authors**
> > >
> > > If the submission is accepted, I hope the authors will really go the extra mile to satisfy the concerns of reviewer 684B to the greatest extent possible. 684B extended good faith to the authors. To not return this good faith would pose a significant reputational hazard -- especially since the community that studies PBSs in IIGs is so small.

---

### Decision · Program_Chairs · 2023-09-21

**Decision:**

Accept (poster)

**Comment:**

This paper presents two novel results on history filtering in imperfect information games:
- Complexity analysis
- A new Gibbs sampler for trick-taking card games along with an empirical analysis

The criticisms of the reviewers came in three main categories:
1. Misrepresentations / omissions of previous work
2. Clarity of presentation, and
3. Unconvincing empirical results.

On the first point, two reviewers brought up several points. One reviewer, 8L8P, who was originally very critical based on these points changed their opinion based on the response of the reviewers and proceeded to champion the paper for acceptance. One of the main reasons I am recommending acceptance is that I am confident that the authors will address the problems identified for the camera-ready copy. I cannot overstate the importance of acting on this point.

On the second point, I believe the presentation problem created by bundling two independent (but obviously related, in my opinion) contributions is overstated. It might feel abrupt on a first read, but I doubt will significantly affect long-term impact. Also, the effect can be mitigated by camera-ready copy. As such, I expect the authors to add a sentence or two to help with the transition from one topic to the other, and maybe a sentence earlier in the paper preparing the reader for the two disjoint content sections.

On the third point, reviewers seem unconvinced by the empirical results because the sampler is not employed in a subgame-solving algorithm or real agent, nor compared to any baseline, i.e. there's no demonstration of improved performance of a Oh Hell agent. I am too a bit disappointed by this, as it's clearly part of the work's motivation. But, the empirical results that are shown are designed to clearly support the practicality of the Gibbs sampler, which it does well. In light of the hardness results, this is good news, and trick-taking card games form a wide class of games that are both played regularly world-wide and have a long history of use within the AI / ML community.